# Research on an Algorithm of Express Parcel Sorting Based on Deeper Learning and Multi-Information Recognition

**DOI:** 10.3390/s22176705

**Published:** 2022-09-05

**Authors:** Xing Xu, Zhenpeng Xue, Yun Zhao

**Affiliations:** 1School of Mechanical and Energy Engineering, Zhejiang University of Science and Technology, Hangzhou 310023, China; 2School of Information and Electronic Engineering, Zhejiang University of Science and Technology, Hangzhou 310023, China

**Keywords:** deeper learning, multiple information fusion, YOLOv4, express sorting, information to identify

## Abstract

With the development of smart logistics, current small distribution centers have begun to use intelligent equipment to indirectly read bar code information on courier sheets to carry out express sorting. However, limited by the cost, most of them choose relatively low-end sorting equipment in a warehouse environment that is complex. This single information identification method leads to a decline in the identification rate of sorting, affecting efficiency of the entire express sorting. Aimed at the above problems, an express recognition method based on deeper learning and multi-information fusion is proposed. The method is mainly aimed at bar code information and three segments of code information on the courier sheet, which is divided into two parts: target information detection and recognition. For the detection of target information, we used a method of deeper learning to detect the target, and to improve speed and precision we designed a target detection network based on the existing YOLOv4 network, Experiments show that the detection accuracy and speed of the redesigned target detection network were much improved. Next for recognition of two kinds of target information we first intercepted the image after positioning and used a ZBAR algorithm to decode the barcode image after interception. The we used Tesseract-OCR technology to identify the intercepted three segments code picture information, and finally output the information in the form of strings. This deeper learning-based multi-information identification method can help logistics centers to accurately obtain express sorting information from the database. The experimental results show that the time to detect a picture was 0.31 s, and the recognition accuracy was 98.5%, which has better robustness and accuracy than single barcode information positioning and recognition alone.

## 1. Introduction

With the rapid development of the world economy, people’s living standards are improving day by day; At the same time, the rapid development of the internet enables more and more consumers to choose convenient online shopping. According to the statistics of the China Post Bureau, the business volume of express service enterprises in China has reached 108.30 billion, up 29.9% year on year, and business revenue has reached 1033.23 billion yuan, up 17.5% year on year. In 2021, because of the increase of express business, this has put existing logistics systems to a huge test. At present, the sorting of express delivery is mainly by a courier sheet that can be divided into automatic sorting, semi-automatic sorting and manual sorting. Automatic sorting is by use of infrared bar code detection based on radio frequency identification (rfid) technology for delivery information [1]. This method is costly, difficult to popularize, and mainly used in large-scale logistics express sorting centers [2]. Semi-automatic sorting involves a semi-automatic sorting machine based on machine vision for sorting. The staff put the courier sheet upward and then put it on a conveyor belt. A camera above the conveyor belt acquires pictures of the courier sheet, and then the central processor identifies the barcode information to generate an electrical signal and control the conveyor belt to send the express to different areas. Manual sorting done directly by a labor force, which is inefficient. With the continuous development of smart logistics and the limitations of cost, most small logistics sorting centers now adopt a semi-automatic sorting method. However, due to the limitations of the sorting environment and equipment, barcodes cannot normally be recognized in the identification process, which requires manual intervention to increase sorting efficiency.

The courier sheet mainly contains a one-dimensional barcode, three segments of express code and the user’s personal information. The one-dimensional bar code is economical, time-dependent and has abundant logistics information. When this semi-automatic sorting method scans the one-dimensional bar code for sorting, the bar code can be disturbed obscured by stains and distortion after layer-by-layer sorting, which seriously affects express sorting. The light in the warehouse, creases and other factors affect detection of the barcode. An express three-segment code is composed of characters with larger fonts, and each segment represents different information, including cities, outlets and salesmen. However, the three-segment code is also easily affected by the characteristics of its surface, resulting in low accuracy in detecting the three-segment code area. Therefore, in the process of express sorting, the positioning of target information is key to sorting an determines its efficiency. Existing sorting methods based on single information on the courier sheet can be divided into two categories. One is a method based on traditional digital image processing. For example, Huang et al. [3] used Halcon visual recognition technology to extract single three-segment code information from the courier sheet with good accuracy but high cost. Weihao et al. [4] used a Hough transform to detect regions containing barcodes. Katona M [5] used algorithms based on morphological operations to detect barcodes, and an improved version [6] used Euclidean distance maps to match barcode candidates. However, such methods are subject to environmental factors and depend on digital image processing, so it is difficult to detect barcodes accurately and efficiently in complex environments. Another method is based on deeper learning [7]. With the continuous development of convolutional neural networks in object detection, these have been widely used in sorting. Zamberletti A et al. [8] first used a deep neural network for barcode detection, but it was not very effective in practice because it was developed using experimental images. Kolekar A and Ren Y [9,10] used the deeper learning detector of SSD for barcode detection, and achieved good performance under a complex background. Li J et al. [11] used the Faster R-CNN network for barcode detection and achieved better detection results with higher accuracy and strong robustness. Z. Pan [12] used a YOLO algorithm for detection of packages when dealing with the problem of express stacking. R. Shashidhar [13] used a YOLOv3 model and OCR to detect and recognize license plates and achieved good results. Methods based on traditional digital processing are easily affected by environmental factors when detecting the target area, which leads to inaccurate detected areas and may cause recognition errors. The use of deeper learning methods to detect the target area has advantages. In image recognition, a convolutional neural network is used to learn various features of the target, which is more robust in target detection. In this regard, considering the complexity of the express sorting environment, we decided to use a deeper learning method to detect the target information. At the same time, we considered that it would be too simple to select only certain information as the sorting information in the real-time sorting process, and the robustness would be poor. In a complex logistics environment, false detection will occur, which will affect sorting efficiency. Therefore, we chose the multi-information method t better guarantee the accuracy of identification.

In summary, we propose a deeper learning-based multi-information fusion method for courier sheet recognition. The method is mainly divided into two stages, one being the positioning of the one-dimensional barcode and the three-segment code on the courier sheet, and the other the decoding of the barcode and the recognition of the three-segment code. As shown in Figure 1, the images are first input into the target detection network for positioning of the two kinds of information. In enable accurate detection in complex environments, we redesigned the key positioning network, which was optimized based on YOLOv4, and ensured the speed and accuracy of the optimized network. The network includes the backbone feature extraction network of YOLOv4, the spatial pooling layer after adding the cross-stage module, the attention module SE, and the use of FPN structure. As a backbone feature extraction network, CSPDarKet53 can ensure accuracy and greatly reduce the number of parameters. A spatial pooling layer with inter-phase modules was used instead of the original spatial pooling layer, which helped to ensure the accuracy and reduce the number of parameters. The attention module SE was added to enhance features that improved the accuracy of detection. Replacing the original structure with an FPN structure effectively reduces the network complexity and parameter number, and the optimized positioning network is much better than the YOLOv4 network. The next step is the recognition of target information. First, the rectangular boxes containing barcode information and three-segment code information are captured, and the pictures containing barcode information are decoded and output by the ZBAR algorithm. The characters of the three-segment code information box are recognized by Tesseract-OCR text recognition. The string information of the two is written into the text so that the express sorting information can be accurately obtained from the database. Our main contributions are as follows.
We propose a multi-information fusion courier sheet recognition method instead of single information target recognition in the process of express sorting to improve the recognition rate of sorting.The YOLOv4 target detection model was optimized for target information positioning. Compared with other detection networks, the performance of courier sheet detection is more powerful.

The rest of this article is as follows. Section 2 describes the target detection network used by the method. Section 3 mainly describes the recognition method of target information. In Section 4, experiments are described to verify the reliability of the detection model and the recognition algorithm, and the whole method is evaluated to verify the feasibility of the proposed method. Section 5 provides the conclusion.

## 2. Target Detection Network

Widely used target detection algorithms in current target detection tasks are all based on deep convolutional neural networks [14] that can learn features from a large amount of data. At present, detection is mainly divided into two-stage with detectors such as R-CNN, Fast R-CNN, Faster R-CNN [15,16,17] and single-stage detectors such as YOLO [18] series and SSD [19]. The output of the single-stage detector only needs a CNN operation to obtain the result directly. The two-stage detector needs to be divided into two steps. The first step is to perform a simple CNN operation, and the second step is to score the results obtained in the first step. Then, the candidate regions with high scores are input into CNN for final prediction. Because of the existence of candidate regions, the two-stage detector has high accuracy but is not as fast as the single-stage detector. Therefore, for fast real-time target detection, a single-stage detector is preferred. Whether target detectors are efficient free (e.g., CenterNet [20]) or anchor based (e.g., EfficientDet and YOLOv4. [21,22]) divides them into two types based on anchor points. The biggest advantage of the former is that the speed of the detector is very fast, and there is no need for preset anchor and direct regression, which greatly reduces time consumption and computational power. The latter has higher accuracy and can extract richer features, but it takes more time and computational power. Therefore, our research considers this selection.

YOLOv4 was improved on the basis of YOLOv3 [23]. As an efficient and powerful target detection model, it takes into account both speed and accuracy. It is mainly composed of three parts: a feature extraction network, Backbone; a Neck for feature fusion, and a detection Head, Yolo Head, for classification and regression operation.

As shown in Figure 2, a picture of the courier sheet captured is input into the YOLOv4 network. The network first adjusts the picture to the size of [3, 416, 416], and then trunk feature extraction network CSPDarknet53 extracts target features. A shallow feature map, deep feature number, and a deep feature map are introduced into the Neck part. After using the SPP structure to enhance the receptive field on the deep feature map, the three feature maps are put into the path aggregation network PANet [24] to extract features repeatedly, and finally into the Yolo Head. The image can be divided into [52, 52, N], [26, 26, N] and [13, 13, N] feature maps of different sizes for detection of large targets, medium targets and small targets, where N=3×(5+C), which depends on the model category.

The loss function of YOLOv4 can be divided into three parts: confidence loss $L_{conf}$, classification loss $l_{class}$, and regression frame loss $l_{CoU}$. $L_{conf}$ and $l_{class}$. These are expressed in Equations (1) and (2).
(1)$$L_{conf}=\overset{s^{2}}{\underset{i=0}{\sum }}\overset{B}{\underset{j=0}{\sum }}I_{ij}^{obj}{\left(C_{i}-{\hat{C}}_{i}\right)}^{2}+\lambda_{noobj}\overset{s^{2}}{\underset{i=0}{\sum }}\overset{B}{\underset{j=0}{\sum }}I_{ij}^{noobj}{\left(C_{i}-{\hat{C}}_{i}\right)}^{2}$$
(2)$$l_{class}=\overset{s^{2}}{\underset{i=0}{\sum }}I_{i}^{obj}\underset{c\in classes}{\sum }{\left(p_{i}\left(c\right)-{\hat{p}}_{i}\left(c\right)\right)}^{2}$$

Regression box loss represents the error between the prediction box and the real box. To ensure more accurate calculation results, several aspects are considered, including the overlapping area of the detection frame, the distance of the center point, and the length-width ratio. Regression box loss $l_{CoU}$ formula is shown in Equation (5).
(3)$$v=\frac{4}{\pi^{2}}{\left(arctan\frac{\omega^{gt}}{h^{gt}}-arctan\frac{\omega }{h}\right)}^{2}$$
(4)$$\alpha =\frac{v}{\left(1-\mathrm{IoU}\right)+V}$$
(5)$$l_{CoU}=1-\mathrm{IoU}+\frac{d^{2}}{c^{2}}+\alpha v$$

Note that α and v are penalty terms for the aspect ratio, $\omega^{gt}$ and $h^{gt}$ are the width and height of the real box, w and h are the width and height of the predicted box, d is the Euclidean distance between the two center points, and c is the diagonal distance of the closure.

The loss function of YOLOv4 is expressed in Equation (6).
(6)$$loss\left(object\right)=l_{CoU}-l_{conf}-l_{class}$$

### 2.1. SPP Module of Csp Modularization 

In deeper learning, the high-level network layer has a large receptive field, so it has a strong ability to represent semantic information. However, the feature map has low resolution and poor ability to represent spatial information. The receptive field ratio of low layer network layer is small, in contrast to that of high layer network layer. Therefore, spatial pyramid pooling SPP [25] was proposed to deal with these problems [26]. This structure is mainly about the maximum pooling of 5 × 5, 9 × 9 and 13 × 13 with different sizes after convolution, batch normalization and activation function. The maximum pooling of the characteristic graph is joined together to change the channel to 2048 with the original size unchanged. Such operation by integrating different receptive fields can enrich the semantic information of feature maps and effectively improve model performance [27]. At the same time, we know that CSPDarknet53, the backbone feature extraction network of YOLOv4, is the key factor in obtaining good results with this network. The cross-stage part network (CSPNet [28]) is a structure proposed from the perspective of network architecture, as shown in Figure 2, CSP_X. This structure divides the input part into two parts, and the backbone part continues the residual The stacking of the other part is directly connected to the end to achieve channel splicing with the backbone part, which is equivalent to a large residual edge. Splitting first and then overlapping greatly reduces the number of parameters and computation, and meanwhile strengthens the CNN’s learning ability and eliminates a computing bottleneck [29]. A K layer CNN with B basic layer channels is shown in Table 1 below.

In addition to the CSP structure of the trunk network, we considered combining the SPP structure mentioned above with the CSP module and optimizing it in the network. This KIND of CSP modular SPP structure reduces the amount of calculation resulting from increasing the SPP module and improves accuracy, achieving the purpose of reducing parameters but ensuring accuracy [28]. The improved CSP-SPP module is shown in Figure 3.

### 2.2. Attention Module SE

The attention model was originally used in machine translation and has become an important part of neural networks. The attentional mechanism module can pick out helpful features by attaching weights to different concerns within the network. Among many attention modules, the SE module is the classic. This focuses on the relationships between channels so that the model learns only useful channel characteristics. It first reduces the dimension of spatial features to 1 × 1 by global average pooling based on the width and height of feature graphs, as shown in Equation (7). Then, two fully connected layers and nonlinear activation functions are used to establish connections between channels, as shown in Equation (8).
(7)$$z_{C}=F_{sq}\left(x_{c}\right)=\frac{1}{1-1\times w}\overset{H}{\underset{i=1}{\sum }}\overset{W}{\underset{j=1}{\sum }}x_{c}\left(i,j\right)$$
(8)$$\hat{z}=T_{2}\left(ReLU\left(T_{1}\left(z\right)\right)\right)$$

The normalized weight is obtained by a Sigmoid activation function, and weighted to each channel of the original feature map by multiplication to complete the re-calibration of the original feature by channel attention, as shown in Equation (9) below.
(9)$$\hat{x}=x\cdot \sigma \left(\hat{z}\right)$$

After global average pooling, the global receptive field can be obtained. During the first full connection, the parameters and calculation amount are greatly reduced by reducing the dimension of the feature graph. Following the nonlinear activation function, the correlation between channels is completed by restoring the original channel number through a full connection. See Figure 4.

### 2.3. Use of the Feature Pyramid Structure

We used a feature pyramid structure, FPN, to replace the PANet path aggregation structure. PANet is an improved version of FPN, which adds a top-down path after a top-down path to achieve feature fusion. Such a structure can be more beneficial to classification and positioning, but at the same time greatly increases the cost of computing. The object features to be detected in our study are not complex, and the difference between the two structures is not obvious. However, it was hoped that the computation and complexity of the network would be reduced, so the FPN structure was used for feature fusion.

### 2.4. Improved YOLOv4 Algorithm

The structure of the detection network is shown in Figure 5. We continued to use the backbone feature extraction network of YOLOv4, and added the SE module after three output layers and after up-sampling to improve positioning accuracy. After that, the backbone part was used with the above-mentioned Csp-spp module to reduce more parameters while improving the receptive field, and finally we used the FPN structure to fuse the features and then output the targe.

## 3. Identification of Target Information

### 3.1. Barcode Decoding

To decode a barcode on the courier sheet, we chose the Zbar algorithm for the decoding operation. The Zbar algorithm is an open-source barcode detection algorithm online. The algorithm can not only read a variety of sources of barcode, such as image files, and videos, but also supports a variety of barcode types, including EAN-13/UPC-A, UPC-E, EAN-8, Code128, Code38, and QR. Our form of bar code was mainly code128. This is shown in Figure 6. Code128 consists of a series of parallel bars and blanks divided from left to right into left margin, start bit, data, validator, end bit, and right margin.

(1) Band Code. Four values of 1, 2, 3 and 4 are assigned according to the thickness, thickness and width of the bar, and the blank respectively. The Band Code of the barcode can be obtained successively.

(2) Left and right-side blank area. A blank space should be left on both sides of the bar code and the width should be 10 times the unit width (note: the unit width is the stripe width of width (1), allowing the bar code reader to enter the readability stage.

(3) Starting bit. The bar and blank detected in the first area of the barcode, which is the beginning of the visible part of the barcode, is composed of six interwoven bars and blanks of different thickness, with a total of 11-unit widths. In Code128, the starting bits of code A, B, and C are 211412, 211214, and 211232 respectively. The type of Code128 is determined by the start bit.

(4) Data. The data area expresses the coding information of the barcode, which is composed of multiple characters. Each character also consists of six bars and blanks.

(5) Validator. This is used to verify the validity of the barcode. The method of checksum module 103 was adopted, and the calculation method [31] is shown in Equation (10).
(10)C=I·Nmod103

N is the value of the bit data.

(6) End character. This indicates the end-state of the barcode, which is fixed, and the corresponding Band Code is 2331112;

After the image is put into the detection network to detect the area of the bar code, the rectangular box containing the bar code must be captured. After the captured image is put into the ZBar algorithm, the algorithm analyzes and scans the image, and determines the Band Code of the bar code by the width of the bar and the empty, to extract the character information contained in the bar code.

As shown in Figure 6, the string “ST089030003” was identified by this algorithm, so that the sorting information of the express could be retrieved from the database.

### 3.2. Recognition of Three Segments of Code

Three-segment code characters are mainly printed bodies combining digits, hyphens and English letters. After obtaining pictures containing three-segment code characters, OCR (Optical Character Recognition) Character Recognition is required. Only when the string information is obtained can the sorting information corresponding to the character be obtained through the database. From the collected data, it was seen that the character distortion of the package would inevitably occur during the transportation process, and the recognition environment could be complex, which would affect the accuracy of recognition. Therefore, Tesseract was used for recognition. Tesseract is an open-source OCR engine. The fourth-generation version can support deep learning OCR, can recognize multiple formats of image files, and convert these to text. Figure 7 shows the single three-segment code style on the express side.

After obtaining the rectangular box containing three sections of code information, we used OpenCV and Tesseract together to obtain text recognition. As shown in Figure 8, after the image is first input, we use OpenCV’s EAST text detector to detect the text in the image. The EAST text detector provides the bounding box coordinates of the text ROI. We extract each text ROI and input these into the LSTM deep learning text recognition algorithm of Tesseract V4. Finally, the output of the LSTM provides the actual OCR result, which is a string. After obtaining the string, we find the sorting information represented by the corresponding number through the database.

## 4. Experimental Design

### 4.1. Dataset 

A dataset was created to simulate the environment of logistics and contained a total of 1680 images, mainly captured by cameras. The pictures of express delivery sheets included multiple express companies and different materials and different sizes, and were sampled under different lighting conditions and different angles. After obtaining the dataset, we used the open-source labeling tool labelimg to label in the VOC dataset format. Before image training, we also augmented the data set, and improved the generalization performance of the model by adjusting the image rotation angle, hue, saturation and other operations. Finally, the data set was divided into a training set and a test set with a ratio of 9:1. Each sample corresponded to two files, namely (1) a JPG file with the image of the package containing the express receipt, and (2) an xml file that stores image information, labels and coordinates corresponding to the region of interest in the image.

### 4.2. Experimental Environment and Training Process

This experiment used the operating system win 10 64 and the neural network framework pytorch. The hardware configuration included a CPU with Intel(R) Core (TM) i9- 10900K CPU @ 3.70 GHz 3.70 GHz; RAM is 64 GB; GPU is NVIDIA GeForce RTX 2080 Ti.

In the object detection network experiment, the size of the input image was 416 × 416, the batch size was 16, the maximum number of iterations was 100, the initial learning rate was 0.001, and the attenuation coefficient was 0.0005. The ratio of training set to test set was 9:1.

At the same time, using the pre-trained model in the detection network, an accurate model could be obtained in a short time by transfer learning.

### 4.3. Analysis of Detection Experiment Results

#### 4.3.1. Evaluation Index of Experimental Results

The target detection model was applied to the distribution center for real-time detection, so the detection speed and accuracy were more important evaluation criteria. The experiment used frame rate per second (FPS) as the speed evaluation index. The FPS value reflects the number of pictures that can be processed per second. The higher the FPS, the faster the detection speed. After that, the FPS data was obtained in the above configuration. Finally, it was decided to use the average precision (*AP*), precision rate (*P*), recall rate (*R*), F1-measure (*F* 1) value, model size and FPS in the detection network to evaluate the network performance. The calculation formulas of P, AP and F1 are expressed as Equations (10)–(12):(11)$$P=\frac{TP}{TP+FP}$$
(12)$$AP=\int_{0}^{1}P\left(R\right)d_{R}$$
(13)$$F1=\frac{2TP}{2TP+FP+FN}$$

Among them, TP represents positive samples predicted to be positive, FP represents negative samples predicted to be positive, and FN represents positive samples predicted to be negative.

#### 4.3.2. Improved YOLOv4 Model Evaluation

The results of the improved YOLOv4 model are shown in Figure 9. It can be seen from various indicators that the experimental results of this positioning network model were good.

##### Ablation Experiments

In this section, the SPP structure combined with CSP structure is denoted as CS-YOLOv4, and the SE module is denoted as SCS-YOLOv4. Through experimental testing, we found that the performance of our model was improved in various aspects.

It can be seen from Table 2 and Table 3 that the optimized YOLOv4 network SCS-YOLOv4 has different degrees of improvement in AP, P, FPS and size compared with the YOLOv4 network. In particular, in the detection of three-segment codes, the AP value increased by 1.7 percentage points, and the *p* value increased by 3.5 percentage points. 

##### Comparative Experiments of Different Models

In our study, the common positioning model and SCS-YOLOv4 model were selected to compare their performance. Table 4 and Table 5 show model comparisons with respect to five aspects of AP, F1, P, FPS and size. All the results were obtained from the same data set.

The two tables above clearly show the differences between the models. The two-stage target detection network Faster R-CNN model has a significant advantage in accuracy, but the model detection speed is too slow and the model is too large. The SSD300 test model has faster speed and more suitable size, but the accuracy is slightly different from other models. YOLOv4 network takes into account both speed and accuracy, and performs well as a whole. Considering that speed and accuracy are important indicators for sorting of express deliveries in a logistics center, and our network is more powerful than the YOLOv4 network and has been improved in various aspects, we used our network to achieve the positioning of target information.

### 4.4. Experimental Results Analysis

After the SCS-YOLOV4 algorithm was used to complete positioning, information was identified. During recognition, we found that it was difficult to accurately and quickly identify a picture with a large deflection angle, so it was necessary to correct this. We intercepted the extent of the bounding box, then the edge detection algorithm in OpenCV is used to process the captured image, and the minAreaRect () method was used to obtain the deflection angle of the image. Finally, affine transformation was used to correct the deflection image. After the corrected picture was obtained, we used the ZBar algorithm to identify the bar code, and the Tesseract to identify the three-segment code. 

#### 4.4.1. Evaluation of Experimental Results

In the logistics environment, if you want to carry out real-time recognition, recognition accuracy P and recognition speed S are important indicators. P is defined in Equation (14).
(14)$$p=\frac{N_{1}}{N}\times 100\%$$
where N is the number of samples, and $N_{1}$ is the number of correctly identified samples.

##### Barcode Decoding Test

We selected 200 bar code pictures as samples for the bar code recognition experiment. The experimental results are shown in Table 6.

##### Three-Segment Code Identification Test

We selected 200 images of three sections of code as samples for the barcode recognition experiment. We found that the image had a lot of interference information which affected recognition accuracy. Therefore, after rotation correction, three code regions were positioned to reduce the interference information and increase the recognition accuracy. The experimental results are shown in Table 7.

##### Multi-Information Target Recognition Test

We analyzed the recognition results of 200 pictures. From the perspective of the overall method, the success rate of express sorting recognition was 98.5%, whether it was single information recognition or multiple information recognition. The results are shown in Table 8.

### 4.5. Time Performance

In our research, the above three algorithm modules were tested separately using 16GB RAM on a 64-bit Windows operating system, with an Intel(R) Core (TM) I7-10875H CPU @ 2.30 GHz, and a main frequency of 2.30 GHz. The running time is shown in Table 7 in seconds (s), and the average running time was 0.31 s when processing a 416 × 416 size package image, as shown in Table 9.

### 4.6. Comparison of Different Express Sorting Methods

We compared several express sorting methods as shown in Table 10.

It can be seen from Table 9 that the method of Liu W et al. is better in time performance, and can reach 0.11 s, but the recognition accuracy is low. The time performance of our method is not outstanding, only 0.31 s, but our method can attain 98.5% accuracy, which is more robust than other methods.

## 5. Conclusions

Aiming at the problem of the low recognition rate of single information sorting methods in small semi-automatic sorting centers, our research proposes a fast recognition method for courier sheet analysis based on deeper learning using multi-information fusion of a one-dimensional barcode and three-segment code. The experimental results show that the method can obtain the information on the courier sheet accurately. At the same time, considering that the overall recognition time is slower than that of single information, we can take the barcode information as the main information and the three-segment code information as the auxiliary information. Only when the barcode cannot be identified is the three-segment code information identified to reduce the sorting time. In general, although this method is slower than the single information recognition method, the multi-information recognition method ensures the accuracy of recognition and has good robustness.

## Figures and Tables

**Figure 1 sensors-22-06705-f001:**
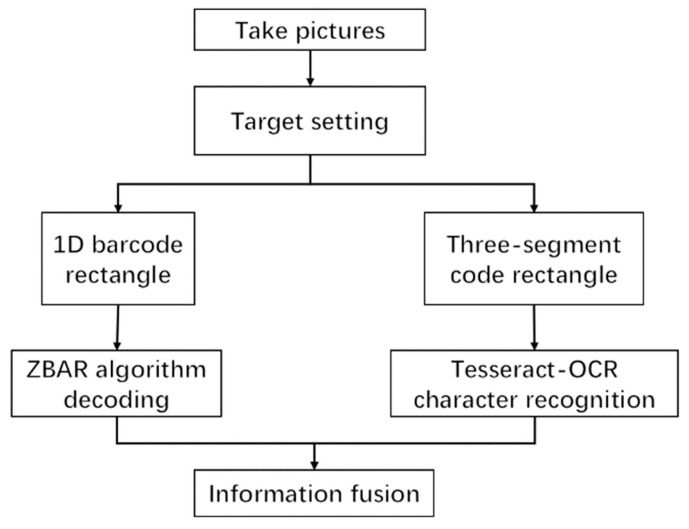
Method structure diagram.

**Figure 2 sensors-22-06705-f002:**
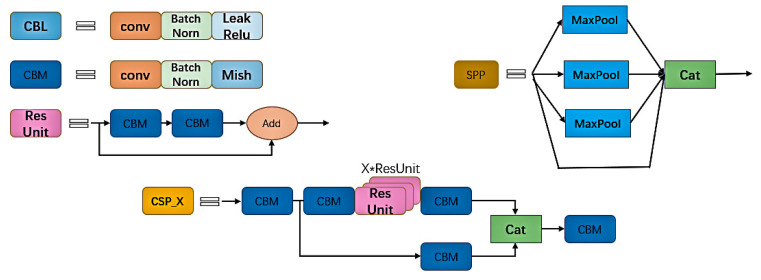
YOLOv4 network.

**Figure 3 sensors-22-06705-f003:**
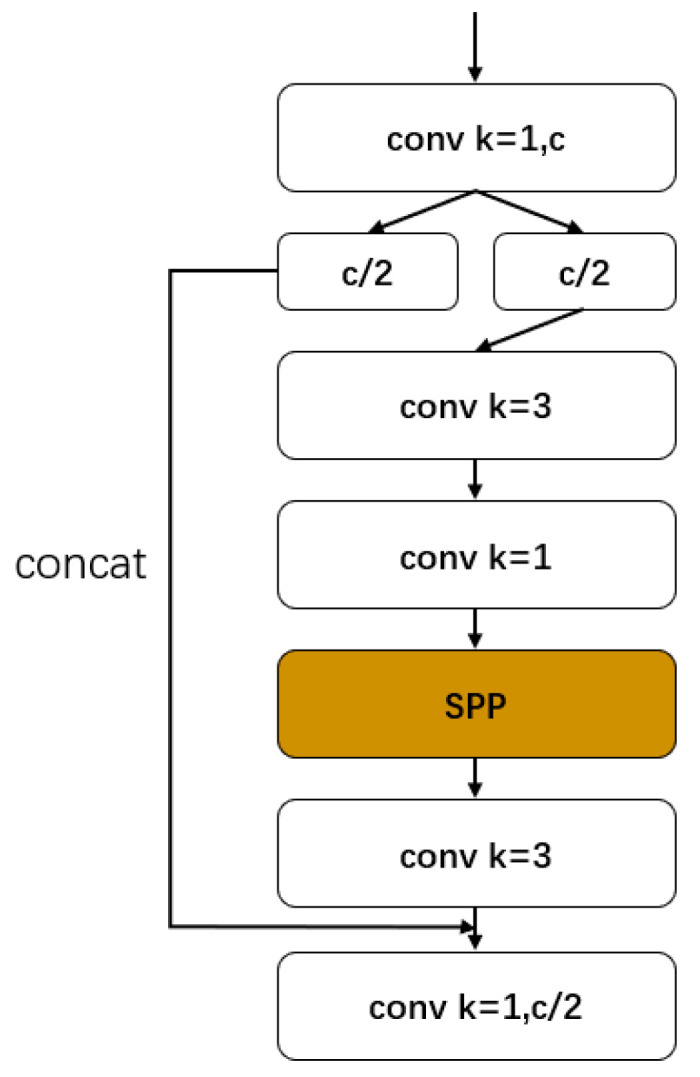
Csp-SPP module.

**Figure 4 sensors-22-06705-f004:**
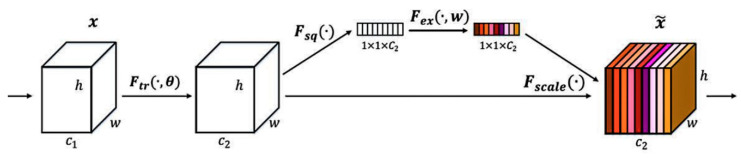
Attention module [30].

**Figure 5 sensors-22-06705-f005:**
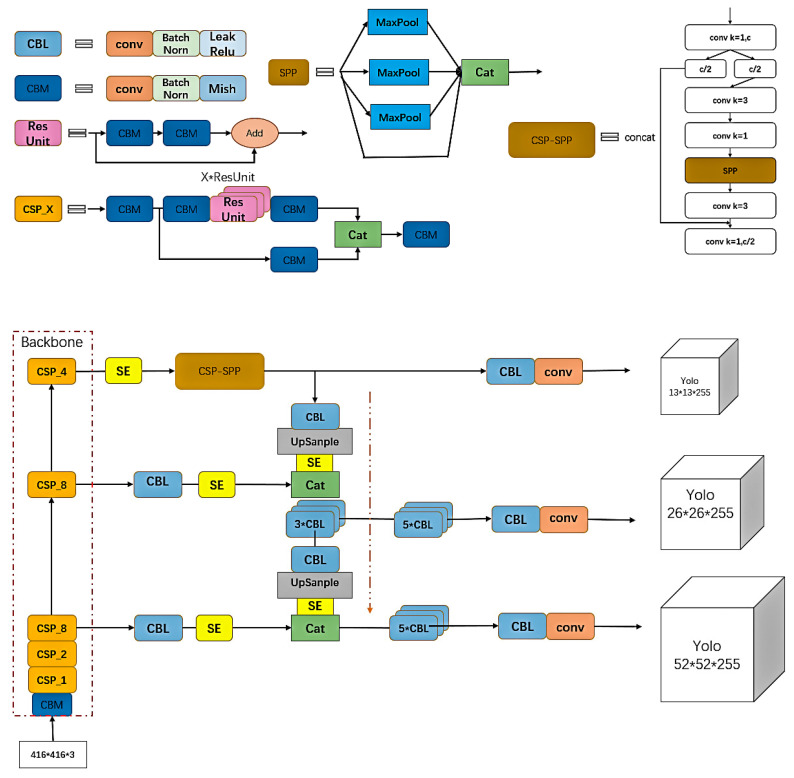
Improved YOLOv4 network.

**Figure 6 sensors-22-06705-f006:**
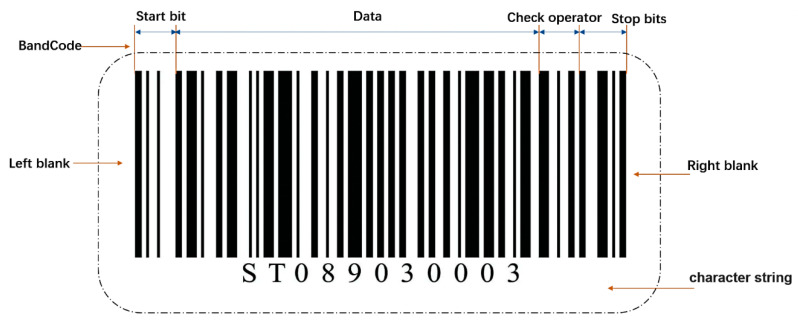
Diagram of bar code.

**Figure 7 sensors-22-06705-f007:**
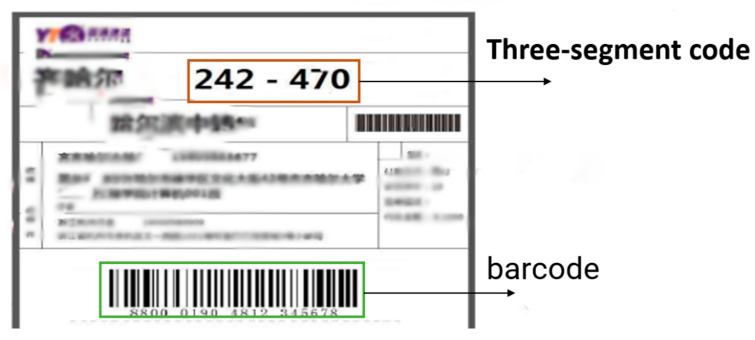
Courier sheet three-segment code style.

**Figure 8 sensors-22-06705-f008:**
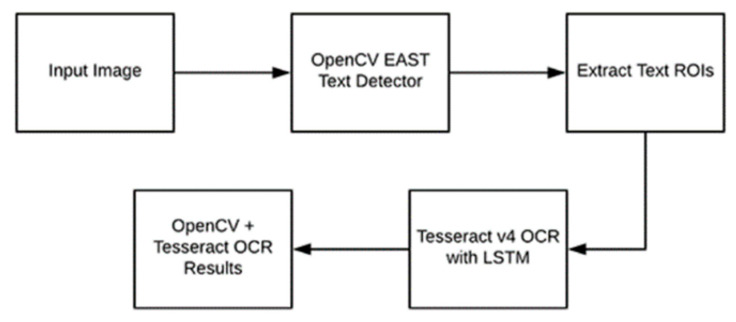
OpenCV OCR flow chart.

**Figure 9 sensors-22-06705-f009:**
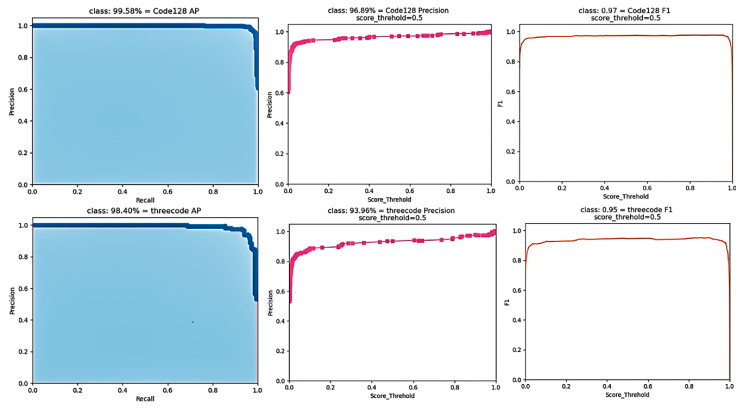
Model result diagram.

**Table 1 sensors-22-06705-t001:** Add CSP structured Dark Layer FLOPs.

Model	Original	To CSP
Dark layer	5whkb2	whb2(3/4+5k/2)

**Table 2 sensors-22-06705-t002:** Comparison of model performance after optimization (bar code).

Model	P	FPS	Size (MB)
YOLOv4	96.6%	21	245
CS-YOLOv4	96.3%	22	174
SCS-YOLOv4	96.9%	26	174

**Table 3 sensors-22-06705-t003:** Comparison of model performance after optimization (three-segment code).

Model	P	FPS	Size (MB)
YOLOv4	73.6%	21	245
CS-YOLOv4	92.4%	22	174
SCS-YOLOv4	94.9%	26	174

**Table 4 sensors-22-06705-t004:** Performance comparison of different models (bar code).

Model	AP	F1	P	FPS	Size
SSD	96.1%	0.86	96.4%	70	110
YOLOv3	85.3%	0.78	87.9%	34	236
YOLOv4	96.9%	0.97	91.44%	21	245
Faster R-CNN	99.8%	0.99	99.9%	12	522
Ours	99.6%	0.97	96.9%	26	174

**Table 5 sensors-22-06705-t005:** Performance comparison of different models (three-segment code).

Model	AP	F1	P	FPS	Size
SSD	86.5%	0.66	94.4%	70	110
YOLOv3	85.3%	0.78	87.8%	34	236
YOLOv4	96.7%	0.93	91.4%	21	245
Faster R-CNN	99.0%	0.98	95.7%	12	522
Ours	98.40%	0.95	93.9%	26	174

**Table 6 sensors-22-06705-t006:** Barcode decoding test results.

The Number of Samples	Correct Number of Decoders	Decoding Success Rate
200	194	97%

**Table 7 sensors-22-06705-t007:** Barcode decoding test results.

The Number of Samples	Correct Number of Decoders	Decoding Success Rate
200	192	96%

**Table 8 sensors-22-06705-t008:** Method identification success rate.

The Number of Samples	Number of Correct Output Messages	Decoding Success Rate
200	197	98.5%

**Table 9 sensors-22-06705-t009:** Method time performance.

Object Detection	Decoding of Bar Code	Three-Segment Code Identification
0.04	0.08	0.19

**Table 10 sensors-22-06705-t010:** Different Identification methods of the courier sheet.

Method	Target Detection Time	Total Time	Information Recognition Accuracy
Katona M [5]	0.05	0.15	91.6%
Liu W [8]	0.02	0.11	93.6%
Polat E [32]	0.48	0.57	97.4%
Ours	0.04	0.31	98.5%

## Data Availability

Data available on request due to restrictions e.g., privacy or ethical. The data presented in this study are available on request from the corresponding author. The data are not publicly available due to our dataset is about courier package, there is a lot of personal information on it.

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
