# Peer review of "Research on an Algorithm of Express Parcel Sorting Based on Deeper Learning and Multi-Information Recognition"

_sensors, 2022, doi:10.3390/s22176705_

Round 1
Reviewer 1 Report
In this paper, the author focuses on the extraction of more robust features from barcodes and three-segment codes to improve the efficiency of express sorting. The guiding idea of ​​this paper is to mitigate the influence of the environment on information extraction through information fusion. To achieve this purpose, this paper fuses single barcode and three-segment code to extracts richer and more robust features from the fused various information. At the same time, this paper makes improvements to YOLOv4, adding SPP module and SE module on it. The SPP module combines deep features with rich semantic information but low resolution and shallow features with less semantic information but high resolution to make the extracted features more discriminative. SE attention can enhance important features to make the final features more accurate. Although this paper analyzes the problems existing in the current logistics sorting system and makes some improvements, the work done in this paper is basically the same as the previous works. Overall, the design of the work done in this paper is relatively simple, and the innovation is not enough. In addition, this paper also has the following problems:
(1) There are many grammatical errors, such as: “this way the high cost is difficult to popularize” “staff need to express the single up after put on the conveyor belt”
(2) The main purpose of this paper is to perform information fusion, but lack of fusion steps, just simply input two different information into the network at the same time.
(3) The SPP module added in this article is already exists in YOLOv4.
(4) The SE attention added in this paper has been used by other object detection works.
(5) This article analyzes the advantages and disadvantages of different detectors, but does not summarize the application scenarios of this article, and fails to summarize the reasons for choosing YOLOv4.
Reviewer 2 Report
The work is well structured and English language looks quite good. The article is written intelligibly in a good scientific style and is easy to read.
However, there are some issues at the paper, which should be addressed: Please, revised the word, which you used into the text for mathematical expression – Formula or Equation. My personal opinion is that equation is more appropriate. Please improve the quality of the figures 2 (upper one), 4, 5, 6 and 9 – the text is blurred.
Reviewer 3 Report
Comment 1:
Motivation of the research reported in this paper is not strong. The need of deep learning is not well discussed. What problem would be without using the deep learning approach?
Comment 2:
There are numerous English writing errors, eg,
- ”whole express sorting”.
- “Method of express surface”.
- “Target information localization”.
- In literature, remove [], etc.
Comment 3:
The definition used in this paper is not quite correct, see the discussion of the definition of deep learning in literature “on definition of deep learning” (2018 World Automation Congress (WAC), 2018, IEEE).
Comment 4:
Definition of single recognition method needs to be given first. The terminology is quite weird; it never appears in the field of manufacturing or service informatics.
Definition of multi-information wrapping surface needs to be given as well.
Comment 5:
How about the robustness of the method due to imprecise information, such as uncertainty, ambiguity, and missing according to the classification in literature. Perhaps, authors want to address this problem in the future work.
Comment 6:
How to determine the weights when fusing multi-pieces of information?
Subjectively determined?
Reviewer 4 Report
Dear Authors,
I have gone through the manuscript titled "Design of a single recognition method for multi-information wrapping surface based on deep learning ". The paper talks about an improved approach for rapid barcode recognition system. Although, this topic is of interest given the potential implication of improvement in sorting process, I fail to identify the impact of novelty of the manuscript. Sorting using barcode is not new in itself. Here authors have used deep learning method to help improve the sorting using barcode recognition. Author has not given a lot of details about their training databases. Also authors have assumed that the barcode remain purely intact in original form. Often during transportation there are scratches, overmarks, imprints on barcode surface, which may affect the results of their method. None of this has been considered in the study.
Also, I feel that title of the paper does not fully reflect the content of the manuscript. Its should be revised. Novelty of the method proposed should be brought out. Perhaps by adding a section on novelty of the method. Additionally, a table should be included where comparison with other barcode recognition methods should be provided to truly highlight the importance and novelty of method proposed.
Figure 7 and 9 should be improved they are of very poor resolution.
I will suggest paper be accepted for publication if the above issues are resolved.
Thanks
Round 2
Reviewer 1 Report
No
Reviewer 3 Report
I am fine with the revision.